# OpenReview forum: "Large Language Models as Automated Aligners for  benchmarking  Vision-Language Models"
_ICLR.cc/2024/Conference — ICLR 2024 poster_

### Official Review · Reviewer_G18f · 2023-10-28

**Soundness:** 2 fair
**Presentation:** 3 good
**Contribution:** 2 fair
**Rating:** 5
**Confidence:** 4

**Summary:**

The paper introduces Auto-Bench, an automated benchmarking pipeline that utilizes Large Language Models (LLMs) to curate data and evaluate Vision-Language Models (VLMs). The pipeline includes LLMs as automatic curators to generate question-answer-reasoning triplets and LLMs as judges to assess VLMs' performance. The paper shows the effectiveness of Auto-Bench in data curation, model evaluation, and supervised fine-tuning of VLMs.

**Strengths:**

1. The paper introduces an innovative and comprehensive benchmarking pipeline for VLMs.
2. The use of LLMs as automatic curators and judges adds scalability to the evaluation process.
3. The extensive dataset curated by Auto-Bench enables effective evaluation and fine-tuning of VLMs.

**Weaknesses:**

1. From my perspective, the approach provided by Visual Instruction Tuning [1] for constructing multimodal instruction data also constitutes a scalable, user-friendly, and comprehensive automated pipeline. Therefore, the claim about data generation is relatively weak. I've noticed that you mention your method being more diverse and complex, but in the analysis of 3.3 BENCHMARK STATISTICS, there is no mention of Visual Instruction Tuning. Lastly, could you please elaborate on any other significant differences and advantages of your method against Visual Instruction Tuning?
2. The use of GPT-4 or GPT-3.5 has already been mentioned in other articles, such as [2] and [3].
3. Assessments of different visual capabilities have also been addressed in other evaluations, such as [2], [3], [4], and [5].
[1] Visual Instruction Tuning
[2] MM-Vet: Evaluating Large Multimodal Models for Integrated Capabilities
[3] TouchStone: Evaluating Vision-Language Models by Language Models
[4] MME: A Comprehensive Evaluation Benchmark for Multimodal Large Language Models
[5] MMBench: Is Your Multi-modal Model an All-around Player?

**Questions:**

Please check the weakness.

---

> ### Author Response · Authors · 2023-11-21
> **Responses to Reviewer G18f (Part I)**
>
> Dear Review G18f,
>
> Thank you for your valuable feedback and comments on our paper. We have carefully considered the raised concerns and address them accordingly in the following.
>
> > **Q1: Comparisons with Visual Instruction Tuning**
>
> We appreciate the reviewer's observation on Visual Instruction Tuning. While we aggree that both our methods and Visual Instruction Tuning utilize LLMs to generate data automatically, our methods offer some significant differences as what follows:
>
> (1) **Diversity on the ability categories**
> The data generated by Visual Instruction Tuning does not specify the abilities that need to be evaluated. The questions they generate only broadly cover five aspects: object types, counting the objects, object actions, object locations, and relative positions between objects. In contrast, our generation is ability-aware, where we have specific prompts for each ability to generate content that targets each skill individually. For the final generated data, **we have covered a total of 17 different abilities (which is more than 4 times of Visual Instruction Tuning)**. Therefore, our generated data exhibits higher diversity in terms of ability categories.
>
> (2) **Diversity on the visual symbolic information**
> Visual Instruction Tuning provides only limited caption and 2D bounding box information to prompt GPT in generating questions. This, to some extent, restricts the diversity of generated questions. For instance, its visual symbolic information lacks OCR information, resulting in the absence of OCR-related questions in its generated data. In contrast, our approach incorporates richer visual symbolic information, such as text, caption, instance relationships, and etc. Therefore, we can prompt GPT to generate data that covers a wider range of information.
>
>
>
>
> > **Q2: Utilization of GPT**
>
>
> Thanks for highlighting the concurrent works [2] and [3]. While these referenced works also employ GPT for benchmark evaluation, notable distinctions exist between our approach and theirs, as outlined below.
>
> (1) **The way of leveraging GPT is different.**
> Concurrent works (e.g., [2] and [3]) solely employ GPT for open-ended evaluation, while our approach integrates GPT not only in automatic evaluation but also in the data curation process. Specifically, we utilized GPT to automatically generate ability-aware questions, spanning a total of 17 different abilities. In contrast, [2] and [3] rely on existing datasets for evaluation, which, to some extent, may lack comprehensiveness. For instance, [2] comprises only 187 images and 205 questions, posing challenges for a comprehensive evaluation. In our exploration, beyond conducting evaluations based on GPT, we also showcased the advantages of using GPT as data curators, highlighting the potential of GPT in this context.
>
> (2) **The purpose of leveraging GPT is different.** [2] and [3] utilized GPT-4 primarily for automated assessment. In contrast, we aim to present a system employing GPT for both data generation and evaluation, thereby offering a fully automated, scalable, user-friendly, and comprehensive pipeline

---

> ### Author Response · Authors · 2023-11-21
> **Responses to Reviewer G18f (Part II)**
>
> > **Q3: Assessments of different visual capabilities**
>
> We acknowledge that assessments of different visual capabilities have been mentioned in previous methods [2, 3, 4, 5].  Therefore, we conducted a quantitative comparison based on the number of assessed capabilities and the quantity of evaluation questions to highlight the diversity of our approach. The table below illustrates that our method significantly surpasses previous methods in the number of evaluation questions. For instance, we exceed the evaluation dataset of MMBench by tenfold (28,500 v.s. 2,974). Additionally, our method is comparable to previous methods in the number of assessed capabilities. We can also extend LLM to assess additional capabilities, thereby scaling our dataset further.
>
> | Methods        | MM-Vet[2] | TouchStone[3] | MME[4] | MMBench[5] | VisIT-Bench[6] |AutoBench(ours) |
> | -------------- | -- | -- | -- | -- | -- | -- |
> | Ability Amount | 6 | 27 | 14 | 20 | 70 | 17 |
> | QA Pairs Amount | 205 | 908 | 1477 | 2974 | 592 | 28500 |
>
>
> We sincerely thank you for your time! Hope we have addressed your concerns. We look forward to your reply and further discussions, thanks!
>
> Sincerely,
>
> Authors
>
>
> [1] Haotian Liu, et al. Visual Instruction Tuning. arXiv preprint arXiv:2304.08485, 2023.
>
> [2] Weihao Yu, et al. MM-Vet: Evaluating Large Multimodal Models for Integrated Capabilities. arXiv preprint arXiv:2308.02490, 2023.
>
> [3] Shuai Bai, et al. TouchStone: Evaluating Vision-Language Models by Language Models. arXiv preprint arXiv:2308.16890, 2023.
>
> [4] Chaoyou Fu, et al. MME: A Comprehensive Evaluation Benchmark for Multimodal Large Language Models. arXiv preprint arXiv:2306.13394, 2023.
>
> [5] Yuan Liu, et al. MMBench: Is Your Multi-modal Model an All-around Player? arXiv preprint arXiv:2307.06281, 2023.
>
> [6] Yonatan Bitton, et al. VisIT-Bench: A Benchmark for Vision-Language Instruction Following Inspired by Real-World Use. arXiv preprint arXiv:2308.06595. 2023.

---

> > ### Author Response · Authors · 2023-11-23
> > **Responses to Reviewer G18f (Part III)**
> >
> > Dear Reviewer G18f,
> >
> > Thanks again for your valuable comments and suggestions in your initial review. Regarding your main concerns on the notations in concept shift, we have revised the paper accordingly and clarified the description in our response. We have also conducted substantial experiments and also revised the paper heavily to make further improvements. Could you please check at your earliest convenience? Thanks!
> >
> > Since the discussion period is approaching the end, we sincerely hope you can let us know if we have addressed your concern and consider raising your score if we do. Also, we welcome any additional discussions or feedback even after the reviewing DDL. Thank you!
> >
> > Sincerely,
> >
> > Authors

---

### Official Review · Reviewer_69zX · 2023-10-31

**Soundness:** 2 fair
**Presentation:** 3 good
**Contribution:** 2 fair
**Rating:** 5
**Confidence:** 5

**Summary:**

The paper proposes AutoBench, a method to generate a large amount of instruction-response pairs for evaluating large multimodal models. The main motivation of the work is centered around the fact that the current datasets are either not suitable for open-ended evaluation or rely on expensive and limited human evaluation. The approach involves using LLM as data creators + LLM as evaluators. I like the motivation of the work, however, the paper majorly lacks in the quality of the experiments and its setup.

Comments:

- My major issue with the setup is using LLMs as data curators! Many existing LMMs are trained on GPT-4 generated data such as LLaVA, mPLUG-Owl etc. To what extent is the evaluation fair across all the models which are not specifically not trained on the GPT-4 generated data?
- The experiments are performed by generating instructions and responses for the COCO images and their corresponding COCO captions. I find this setup highly concerning since InstructBLIP (best model in their eval) is trained with COCO captions. Similarly, other models like LLaVA are also trained with the COCO captions. Again, this makes the evaluation unfair to all the models.
- Quoting from the paper: “ Besides, we carefully curated approximately 28.5K high-quality samples to comprise the validation dataset, which was subsequently utilized for performance evaluation.” There is no information on how this curation was performed?
- Benchmark scope: It remains unclear to me how an image in the dataset is assigned a particular subskill? When do you know that a particular image has physics related possible questions?
- Quoting from the paper: “Due to the nature of reasoning-based questions often lacking a unique answer, we format them as multiple-choice questions, thus reducing the difficulty of evaluation and ensuring its accuracy.” It would be better to provide some examples to understand why reasoning-based questions need multiple choice questions. Why can’t they be just open-ended even if they lack a unique answer.
- Thanks for showing distributions of the question length and cosine similarity. Figure 3b looks more or less overlapping and hence does not shout semantic richness in comparison to other dataset.
- Figure 3 should also have datasets like Visit-Bench or Touchstone where the questions are collected from humans. I feel that the human questions will be the most diverse.
- How much does it cost to add a new model to your benchmark involving 28.5K instances?
- To establish LLM as a valid judge, the paper does not have any correlations with human numbers. In addition, it would be important to show that simpler metrics that are based on lexical similarity are not good judges for this dataset, which it currently does not do.
- I like the paper’s diversification into understanding the usefulness of the models along different skills and sub-skills.

**Strengths:**

Mentioned in the summary

**Weaknesses:**

Mentioned in the summary

**Questions:**

Mentioned in the summary

---

> ### Author Response · Authors · 2023-11-21
> **Responses to Reviewer 69zX  (Part I)**
>
> Dear Review 69zX,
>
> We appreciate your insightful and constructive feedback. We address your comments below, and discuss the revisions we made accordingly.
>
>
> > **Q1: Fairness issue in evaluating VLMs that trained on other GPT-4 generated data.**
>
> While both the Auto-Bench and prior datasets (e.g., LLava-i-158K [0]) are curated using GPT-4, distinct variations exist as shown in the following, leading to a domain disparity between our data and that of others, thus ensuring fair evalution.
>
> - **Auto-Bench incoporates more visual symbolic information.**
> Auto-Bench integrates a more comprehensive set of visual symbolic cues, including OCR, captions, instance relationships, and others. This diversification marks a departure from existing GPT-4 systhesis data.
>
> - **Auto-Bench generates a more diverse set of ability types.**
> Previous benchmark generates questions from limited aspects: object types, object counting, object actions, object locations, and relative positions between objects. In contrast, the dataset in Auto-Bench encompasses a total of 17 distinct abilities, surpassing LLava-i-158K by more than fourfold, for example.
>
> Besides, we observed that not only LLAVA but also most of the models (except BLIP2) compared in the main paper are trained with GPT4-generated data. However, Table 8 indicates there is no strong correlations bwteen the performance and the GPT4-generated data. For example, Blip2 (trained without GPT-4 data) perform better than MiniGPT-4 (trained with limited CC+ChatGPT instruction data) across various sub-skills.
>
> These findings affirm the fairness and balance of our comparisons, addressing concerns about data homogeneity and model bias.
>
> > **Q2: Fairness issue in evaluation VLMs that trained on coco caption data**
>
> Thanks for pointing out this concern. We believe the evaluation is fair due to the following two reasons:
>
> - Compared with COCO caption dataset, Auto-Bench offers **unique and previously unseen** QAs by prompting GPT4 from diverse visual symbolic information. Though sharing the same COCO images, these unsceen QAs and domain disparities ensure the fair evaluation.
>
> - We found that InstructBLIP is not the sole model trained with COCO captions; all models, with the exception of MiniGPT4, underwent training on COCO captions. Despite this shared training data, we did not observe an obvious positive correlation in performance when comparing models trained with and without COCO captions. This finding also underscores the distinctive and complex nature of the Auto-Bench dataset and indicates the fair evaluation.
>
> We greatly appreciate your suggestions of Q1 and Q2 and have thoroughly discussed these potential issues in **Section A.1.5**.
>
>
> > **Q3: Human verfication in validation data**
>
>
> Actually, we employed a crowdsourcing approach for manual validation, assessing each question against three criteria, including (1) the relevance to the sub-skill topic, (2) appropriateness of the answer, and (3) logical consistency in reasoning. The question is deemed valid only when it meets all criteria. **Table 2** in the original main paper presents review statistics for 800 questions across 16 skills. We appreciate your reminder and have updated these details in **Section 3.1** of the revised main paper.
>
>
> > **Q4: How are images in the dataset categorized by specific subskills?**
>
>
> Thanks for the good questions. For general subskills, we designed questions for all images. For specific skills, we filtered images using COCO labels to ensure scenarios matching the skill. For instance, for physics, we only selected images with people and objects; for biology, we chose images with plants, vegetables, fruits, etc. This case-by-case approach also explains why there are fewer questions generated in the reasoning category. We have updated the details in **Section A.1.1** of the Appendix.
>
>
> > **Q5: Some illustrations of Open-ended questions**
>
> We provide the followig case for a better illustration.
> - *Image context*: A downtown street with people hurrying by has a small store with colorful handicrafts in the window. A young man in sneakers and a sweatshirt stops and gazes at the window.
> - *Question related to reasoning*: Why the man gazes at the window of the store?
> - *Answer 1*: The young man is an art lover who is looking for a unique craft to decorate his home and highlight his taste in art.
> - *Answer 2*: The young man, who is actually a basketball player, saw some unique handcrafted basketballs in a store and he wanted to purchase one as a souvenir.
>
> As we can see, for reasoning tasks, there might be numerous correct answers in an open-ended questions, which challenges the LLM's judgment capability, often leading to additional errors. To mitigate this, we structured the reasoning questions in a multiple-choice format. This approach limits the degrees of freedom, enhancing the stability and reliability of the assessment.

---

> > ### Author Response · Authors · 2023-11-21
> > **Responses to Reviewer 69zX (Part II)**
> >
> > > **Q6: Figure 3b looks more or less overlapping and hence does not “shout” semantic richness in comparison to other dataset?**
> >
> > Thanks for the observation. It should be noted that centerline of the distribution denotes the diversity of the data rather than the overlapped area. Figure 3 illustrates that Auto-Bench exhibits a comparatively lower centerline than other dataset, indicating the diversity of the generated questions. To enhance clarity in the revision, we have taken the following modifications: (1) including the detailed calculation of cosine similarity for each dataset individually, (2) adding inclusion of centerline for clear comparisons, and (3) zooming in on areas around the distribution peak in Figure 3.
> >
> >
> >
> > > **Q7: Compared with more datasets like Visit-Bench and Touchstone in Figure 3.**
> >
> > Thanks for highlighting the concurent works (Visit-Bench [1] and Touchstone [2]). In the revision, we've updated the Figure 3 to include statistics of these sources. Visit-Bench displays the lowest average cosine similarity, indicating high diversity, whereas Touchstone has the highest average, suggesting lower diversity. This was confirmed through data examination from their offcial releasing: https://touchstone-beijing.oss-cn-beijing.aliyuncs.com/dataset/touchstone_20230831.tsv. Importantly, Auto-Bench consistently exhibits a low average cosine similarity in comparison to the other five human-curated datasets. Though, it ranks marginally higher than Visit-Bench, the latter is meticiously curated by human with limitted cases (i.e., 592 questions). Overall, Figure 3 underscores our method's capability to produce large-scale, diverse assessment data, effectively balancing quality with quantity.
> >
> >
> >
> > > **Q8: Cost of evaluating a new model.**
> >
> > For quantitative evaluation, we expedited the process using a multi-threaded GPT-3.5-Turbo approach. Each competency evaluation (around 2000 Q&A pairs) takes about 4 minutes, meaning adding a new model to our comprehensive assessment **takes roughly 60 minutes and costs $15**. For qualitative assessment, our Swiss Wheel ELO system streamlines the process, eliminating the need for one-on-one model comparisons, thus reducing both cost and time. Overall, our assessment methodology is efficient and cost-effective. We have added cost details in **Section A.2.4** of the appendix in the revision.
> >
> >
> > > **Q9: Correlations with human numbers.**
> >
> > Thanks for pointing out the importance of establishing correlations between LLM and human judgments. Actually, in our original main paper, we have already addressed this issue in Section 4.4 and Figure 5. To validate the correctness of our approach, we randomly selected 100 open-set samples and distributed them among trained individuals for verification. The results revealed an average consistency of over 90% between the judgments of GPT and human evaluators.
> >
> > > **Q10: Simpler lexical metrics are not good judge.**
> >
> > Thanks for the suggestion. Simple lexical metric are limited in assessing the correctness of complex answers and aligning with human preferences, as shown in various studies [3,4]. Taking the question-answer in Section B.2 as example, it's clear that complex responses from current VLMs defy easy right/wrong judgment via lexical metrics like n-grams. Additionally, solutions, like hardcoded rules, cannot fully represent human preferences. For example, when evaluating VLM responses to questions with harmful content, it's essential for the VLMs to not only reject the content but also provide a clear rationale for the rejection. Such nuanced judgment is beyond the scope of hard-coded metrics, which typically lack the flexibility to assess complex, context-based decisions.
> >
> > We have updated relevant citations and highligted the shortage in the revision.
> >
> >
> > > **Q11: Detailed evaluation of models**
> >
> > Thank you for appreciating the value of our comprehensive evaluation, which examines the models' performance across various skills and sub-skills.
> >
> >
> > We sincerely thank you for your time. Hope we have addressed your concerns. We look forward to your reply and further discussions, thanks!
> >
> > Sincerely,
> >
> > Authors
> >
> > [0] Haotian Liu, et al. Visual Instruction Tuning. arXiv preprint arXiv:2304.08485, 2023.
> >
> > [1] Bitton Y, Bansal H, Hessel J, et al. Visit-bench: A benchmark for vision-language instruction following inspired by real-world use[J]. arXiv preprint arXiv:2308.06595, 2023.
> >
> > [2] Bai S, Yang S, Bai J, et al. Touchstone: Evaluating vision-language models by language models[J]. arXiv preprint arXiv:2308.16890, 2023.
> >
> > [3] Novikova, J., Dušek, O., Curry, A. C., & Rieser, V. (2017). Why we need new evaluation metrics for NLG. arXiv preprint arXiv:1707.06875.
> >
> > [4] Zheng, L., Chiang, W. L., Sheng, Y., Zhuang, S., Wu, Z., Zhuang, Y., ... & Stoica, I. (2023). Judging LLM-as-a-judge with MT-Bench and Chatbot Arena. arXiv preprint arXiv:2306.05685.

---

> ### Comment · Reviewer_69zX · 2023-11-21
> **Response to Authors**
>
> Hi, thanks for answering my questions thoroughly. I am raising my score to 5 as their response alleviates some of my concerns, and will retain it since the idea of the work is not new to me -- GPT-4 has been used for generating instructions and judging responses in prior works separately if not in tandem.
>
> I am adding more comments to improve your current draft.
>
> 1. **Fairness**
>
> - I thank the authors for clarifications on the fairness of the evaluation. According to the authors, most (if not all) of the models have been trained with the GPT-4 generated instructions, COCO captions, and COCO images. I buy the argument that evaluation dataset is (un)fair to all the models equally.
>
> - I do think that this fairness issue needs to be discussed in the main text (even 2-3 lines is fine) instead of completely relegating it to a small Appendix A.1.5
>
> - GPT-4 generated data training: The authors utilize visual symbolic info, diverse set of ability types which they believe is absent in LLaVA-158K dataset. However, all such arguments make sense, I strong empirical evidence is still missing. I suggest the authors to perform similarity analysis between their dataset (and other LMM evaluation dataset) instructions vs LLaVA-158K instructions. It will give you a good sense of the diversity/difference with existing datasets. It will help you understand the nuances of the dataset better.
>
> - COCO Images: Same as above. You can compare the similarity between your dataset (and other LMM eval) dataset images and LLaVA-158K images to better argue the diversity in your dataset.
>
> 2. **Human verfication in validation data**
>
> - If human verification is done on 800 instances, then why do the authors claim 27.5K highly curated samples?
>
> 3. **How are images in the dataset categorized by specific subskills?**
>
> - The authors should provide the heuristics in greater detail in the revised paper.
>
> 4. **Structure of the reasoning questions**
>
> - I do not fully buy the argument provided for keeping reasoning questions as multiple-choice questions. If the LLM is not good as judge on evaluating the responses for the reasoning questions, it is a drawback that should be discussed in the paper. The whole claim of the paper is that LLM can replace humans as judges. But it seems that it is not always the case.
>
> Good luck to the authors for their hard work!

---

> ### Author Response · Authors · 2023-11-23
> **Responses to Reviewer 69zX II (Part1)**
>
> Thanks for the prompt feedback. We are glad to note that we've resolved some of your concerns, including:
>
> - (1) The evaluation on GPT-generated data is fair;
> - (2) The diverse visual symbolic information and ability types differs Auto-Bench from other datasets.
>
> Since there are still some concerns that have not been fully clarified, we aim to address your remaining comments in the following detailed responses.
>
> > Q1: Discussion of the fairness issue in the main paper
>
> Thanks for the suggestions. We have incorporated the relevant discussions (as shown in the following italicized contents) into the Section 3.4 main of main paper to enhance reader comprehension of fairness issues.
>
> *''We aim to evaluate VLMs on the LLM-curated data using LLM's judgments. It is important to acknowledge a theoretical risk of unbalanced evaluation, given that some current VLMs are trained on LLM-generated datasets. However, considering that the Auto-Bench dataset offers substantially distinct datasets compared to previous ones, encompassing more abilities and diverse input visual symbolic information, and given that the selected VLMs are predominantly trained on LLM-generated data, the comparisons can be deemed fair.''*
>
>
> > Q2: The strong empirical evidence of the difference between datasets
>
> As suggested, we have provided two empirical evidences to conduct the differences/similarities analysis between Auto-Bench and others, which are shown in the following:
>
>
> - (1) We have updated Figure 3 by incorporating the statistics of LLAVA-158K, generated by GPT4 as detailed in [1]. The results indicate that while Auto-Bench and LLAVA-158K share similar question lengths, Auto-Bench demonstrates greater diversity, as evidenced by the right part of Figure 3. We attribute this enhanced diversity in Auto-Bench to the incorporation of diverse visual symbolic information and ability types.
> - (2) In addition to intra-correlations, we have conducted inter-correlations as outlined in the following table. Specifically, we randomly selected 2,000 questions from two distinct datasets and computed cosine similarities on the corresponding features extracted by the Simcse text encoder. Results (last line in the following table) shows that though both are generated from GPT-4, LLAVA-158K is not the semantically closest (since the score are not the highest) in resemblance to our Auto-Bench. The following table and related analysis have been updated in Section A.1.5 of the appendix.
>
>
>
> |                | LLaVA[1]              | VQAv2[2] | VisIT-Bench[3]  | TouchStone[4] | GQA[5]    | OKVQA[6]  | Auto-Bench |
> |----------------|--------------------|---------|--------|------------|--------|--------|-----------|
> | **LLaVA[1]**           | -                  | 0.2270  | 0.2209 | 0.2677     | 0.2352 | 0.2351 | 0.2421    |
> | **VQAv2[2]**         | 0.2270             | -       | 0.1969 | 0.2479     | 0.2613 | 0.2486 | 0.2415    |
> | **VisIT-Bench[3]**           | 0.2209             | 0.1969  | -      | 0.2531     | 0.2108 | 0.2210 | 0.2209    |
> | **TouchStone[4]**     | 0.2677             | 0.2479  | 0.2531 | -          | 0.2549 | 0.2590 | 0.2507    |
> | **GQA[5]**            | 0.2352             | 0.2613  | 0.2108 | 0.2549     | -      | 0.2488 | 0.2540    |
> | **OKVQA[6]**          | 0.2351             | 0.2486  | 0.2210 | 0.2590     | 0.2488 | -      | 0.2378    |
> | **Auto-Bench**      | 0.2421             | 0.2415  | 0.2209 | 0.2507     | 0.2540 | 0.2378 | -         |

---

> > ### Author Response · Authors · 2023-11-23
> > **Responses to Reviewer 69zX II (Part2)**
> >
> > > Q3: Human verfication samples *v.s.* high-quality samples
> >
> > Perhaps there is a misunderstanding regarding the datasets mentioned in our paper. Human verfication samples are different from the high-quality samples. The former is solely designated for validating the quality of the LLM-generated data, whereas the latter is employed for the entire evaluation benchmark. It's crucial to clarify that both the human-verified samples and the curated high-quality samples were evaluated using consistent selection criteria. Specifically, we randomly selected 800 samples from the entire pool of question-answer-reasoning triplets for pilot validation purposes. This method was chosen due to its labor-efficient approach while still ensuring adequate validation coverage, as detailed in Table 2. Subsequently, the high-quality human-verified dataset was carefully compiled using a crowdsourcing approach, adhering to the same selection criteria. This process resulted in a dataset comprising 28.5K high-quality samples.
> >
> >
> >
> >
> >
> > > Q4: How are images in the dataset categorized by specific subskills
> >
> > Thanks for the suggestions. We have added the details on how to filtering images for different capabilities in the revision. Indeed, through empirical observation, we noted a close association between the Question-Answer (QA) data corresponding to specific abilities and the content of the images. For instance, in the context of physics reasoning, QAs generated for images featuring people and objects exhibit greater reasonability. Conversely, for biology and chemistry, QAs generated for images containing plants, vegetables, and fruits demonstrate increased coherence. The following table illustrates the details of image filtering based on labels corresponding to the targeted ability intended for curation.
> >
> >
> > | Sub Skills                        | Filter by Labels |
> > |-----------------------------------|------------------|
> > | Object Counting, Object Categorization, Object Localization, Action Prediction, Action Recognition, Text Recognition, Scene Understanding, Common Prediction, Common Explanatory, Common Counterfactual, Short-term Planning | All              |
> > | Physics, Privacy, Security        | 'Person'         |
> > | Biology                           | 'Person', 'Bird', 'Cat', 'Dog', 'Horse', 'Sheep', 'Cow', 'Elephant', 'Bear', 'Zebra', 'Giraffe', 'Banana', 'Apple', 'Orange', 'Broccoli', 'Carrot', 'Potted Plant' |
> > | Chemistry                         | 'Potted Plant', 'Banana', 'Apple', 'Orange', 'Broccoli', 'Carrot' |
> >
> > We have revised and expanded the discussions in the revised version to enhance clarity.
> >
> > > Q5: More details of the reasoning questions
> >
> > Assessing open-ended answers is challenging, which is not a limitation specific to LLM but also for human beings. In fact, we empirically found that, in many cases, the ambiguity arises inherently from the open-ended answers, as illustrated in the example mentioned - *Image context*: A downtown street with people hurrying by has a small store with colorful handicrafts in the window. A young man in sneakers and a sweatshirt stops and gazes at the window.
> > - *Question related to reasoning*: Why the man gazes at the window of the store?
> > - *Answer 1*: The young man is an art lover who is looking for a unique craft to decorate his home and highlight his taste in art.
> > - *Answer 2*: The young man, who is actually a basketball player, saw some unique handcrafted basketballs in a store and he wanted to purchase one as a souvenir.
> >
> > It's important to note that even for humans, determining the correctness of two answers in such scenarios can be uncertain. Therefore, both LLM and humans may have difficulty making accurate judgments in such cases. This motivated us to design multiple-choice questions. In the case of prompting GPT with multiple-choice questions, we explicitly specify in the prompt to generate options that ensure the uniqueness of the answers, thus facilitating subsequent accurate judgments. We firmly believe that with the adoption of more powerful Language Learning Models (LLMs) like GPT-4 and its successors, the challenges we face will be effectively addressed. Our initiative represents a significant step in harnessing a controllable and robust LLM to evaluate and enhance the capabilities and values of other AI systems.
> >
> > We sincerely thank you for your time. Hope we have addressed your remaining concerns. We also look forward to your reply and further discussions. Finally, Happy Thanksgiving and thanks for enhancing our paper!
> >
> > Best,
> >
> > Authors

---

> > > ### Author Response · Authors · 2023-11-23
> > > **Responses to Reviewer 69zX II (Part3)**
> > >
> > > ### Reference
> > >
> > > - [1] Haotian Liu, et al. Visual Instruction Tuning. arXiv preprint arXiv:2304.08485, 2023.
> > >
> > > - [2] Tsung-Yi Lin, et al. Microsoft coco: Common objects in context. ECCV, 2014.
> > >
> > > - [3] Yonatan Bitton, et al. VisIT-Bench: A Benchmark for Vision-Language Instruction Following Inspired by Real-World Use. arXiv preprint arXiv:2308.06595. 2023.
> > >
> > > - [4] Shuai Bai, et al. TouchStone: Evaluating Vision-Language Models by Language Models. arXiv preprint arXiv:2308.16890, 2023.
> > >
> > > - [5] Hudson Drew A, et al. Gqa: A new dataset for real-world visual reasoning and compositional question answering. CVPR, 2019.
> > >
> > > - [6] Marino Kenneth, et al. Ok-vqa: A visual question answering benchmark requiring external knowledge. CVPR, 2019.

---

### Official Review · Reviewer_m7V9 · 2023-11-02

**Soundness:** 4 excellent
**Presentation:** 4 excellent
**Contribution:** 4 excellent
**Rating:** 8
**Confidence:** 4

**Summary:**

The authors use LLMs to generate and curate a new evaluation benchmark for vision-language models, dubbed Auto-Bench. By conditioned on image verbalizations (such as image captions, object locations, OCR etc) LLMs are used to generate question-answer-rationale triplets to evaluate a wide variety of vision-language capabilities, covering perception, reasoning, planning abilities and alignment with human values. To overcome evaluation bottlenecks such as surface form variation, he authors further propose using LLMs to judge model responses. Finally, the authors benchmark several state-of-the-art VL models using their proposed benchmark.

**Strengths:**

- This work addresses an important challenge -- existing vision-language benchmarks only cover a narrow range of capabilities, and are limited in size due to the difficulty of manual curation.

- The primary contribution of this work is substantial -- a large benchmark containing more than 3 million examples, including a train split and a high-quality human-curated validation set, as well as an evaluation framework using LLMs.

- The authors choose a large breadth of the different capabilities to evaluate VLMs on, including planning and human value alignment which have not been focused on much in prior work. I also like how different tasks were framed as open-ended and close-ended depending on the nature of the task.

- I really like the comparison between AutoBench and existing human-curated VQA datasets in section 4.2.

- The experiments are well done. I especially appreciate the Inclusion of qualitative model comparisons using an ELO system.

- The analysis of alignment between LLM and humans as judges in Section 4.4.

- The paper is well-written and easy to follow for the most part.

Well done!

**Weaknesses:**

- The benchmark is built only on images sourced from MS-COCO. While I understand that this was done due to the richness of image annotations that could be fed to the LLM to generate questions, MS-COCO images also represent a narrow subset of all images which we would like to apply vision-language models to. Vision-language models have been oversaturated on COCO-based benchmarks, so performance on Auto-Bench may not be reflective of performance on other image domains (e.g. VizWiz, or medical images)

**Questions:**

In section 4.2, "Users are guided to rank each sample based on its rationality and level of challenge." what does rationality mean?

---

> ### Author Response · Authors · 2023-11-21
> **Respones to Reviewer m7v9**
>
> Dear Review m7V9,
>
> Thanks for your comments and acknowledgment of our novelty. We provide responses to your concerns below.
>
> > **Q1: The used MS-COCO images may not cover the extensive image domains.**
>
> We appreciate and acknowledge the reviewer's observation regarding the limitations of using the COCO dataset, as it may not cover all image domains, such as medical images. The choice of COCO for evaluation was made because of its rich annotations. However, we would like to emphasize that the contribution of this work extends beyond providing a COCO-based dataset for evaluating VLMs.
>
> The primary contribution, in fact, lies in offering an automatic pipeline that is readily extendable to new data domains and scenarios, such as VizWiz. To achieve this, we can leverage off-the-shelf perception models to obtain visual symbolic representations, as mentioned in our paper. Subsequently, we can conduct data curation and model evaluation tailored to the specific domain. We have incorporated this discussion into the revision outlined in **Section A.1.1** of the Appendix.
>
>
>
> > **Q2: Meaning of "Rationality" in Section 4.2**
>
> *'Rationality'* in this context refers to the logical coherence of a question in relation to the specified curation topic and the context presented within the image. We have added the descriptions in **Section 4.2**, for better clarity. Please check the latest revision.
>
>
> We sincerely thank you for your time. Hope we have addressed your concerns with the above responses. We look forward to your reply and further discussions.
>
> Best,
>
> Authors

---

### Author Response · Authors · 2023-11-21
**Responses to ACs and all reviewers**

We sincerely appreciate all reviewers for the time and efforts in reviewing our paper. We are pleased to note that reviewers have acknowledged the following contributions of our work:
 - **Innovative benchmarking pipeline:** This work addresses a significant challenge by introducing an innovative and comprehensive benchmarking pipeline for evaluating current VLMs. [m7V9,G18f]
 - **The extensive dataset:** The primary contribution of this work is substantial, providing a large benchmark containing over 3 million examples, including a train split and a high-quality human-curated validation set. [m7V9,G18f]
 - **The comprehensive evalution:** The authors have undertaken a comprehensive evaluation by selecting a broad range of different capabilities to assess VLMs, covering a diverse set of skills and sub-skills. [m7V9,69zX,G18f]

We also thank all reviewers for their insightful and constructive suggestions, which help further improve our paper. In addition to the pointwise responses below, we summarize the major revisions in the rebuttal according to reviewers’ suggestions.

- **Clarification on the difference with other methods ([69zX,G18f]):** For better understanding Auto-Bench's contributions, we included detailed comparisons with concurrent works such as Visual Instruct Tuning[1], TouchStone[2], MM-Vet[3], MMBench[4], etc.
- **Extended Experiments ([G18f]):**  For comprehensive comparisons, new experiments have been conducted to incorporate additional statistics of different benchmarks.
- **Updating Manuscript ([m7V9,69zX,G18f]):** As suggested, we added some new discussions and included our revision in the paper **(contents in blue)**.

We hope our pointwise responses below could clarify reviewers’ confusion and alleviate all concerns. We thank all reviewers’ efforts and time again and look forward to any further discussions.

Best,

Authors

[1] Haotian Liu, et al. Visual Instruction Tuning. arXiv preprint arXiv:2304.08485, 2023.

[2] Shuai Bai, et al. TouchStone: Evaluating Vision-Language Models by Language Models. arXiv preprint arXiv:2308.16890, 2023.

[3] Weihao Yu, et al. MM-Vet: Evaluating Large Multimodal Models for Integrated Capabilities. arXiv preprint arXiv:2308.02490, 2023.

[4] Yuan Liu, et al. MMBench: Is Your Multi-modal Model an All-around Player? arXiv preprint arXiv:2307.06281, 2023.

---

### Meta-Review · Area_Chair_aLWz · 2023-12-15

**Metareview:**

This work explores using LLMs for curating evaluation benchmarks for vision-language models and creates a novel approach for this called Auto-Bench. Based on text explanations of the image (captions, object locations) Auto-Bench generates question-answer-reasoning triplets for evaluating different capabilities of VLMs. In addition, this work proposes to use LLMs also as part of computing the evaluation metrics by enabling these models to judge the VLM responses. Extensive experimentation is performed to showcase the utility of the method.

**Justification For Why Not Higher Score:**

As pointed out by reviewers, this method is not fully novel as a number of related works pursue very similar ideas and implementations.

**Justification For Why Not Lower Score:**

Authors provide the largest evaluation benchmark (to date) and provide extensive experiments beyond what is currently available in the literature. I believe the community would benefit from this work being presented at ICLR

---

### Decision · Program_Chairs · 2024-01-16

Accept (poster)